# Front-Vehicle Detection in Video Images Based on Temporal and Spatial Characteristics

**DOI:** 10.3390/s19071728

**Published:** 2019-04-11

**Authors:** Bo Yang, Sheng Zhang, Yan Tian, Bijun Li

**Affiliations:** 1State Key Laboratory of Information Engineering in Surveying, Mapping, and Remote Sensing, Wuhan University, Wuhan 430079, China; 13910700045@139.com; 2School of Electronic Information and Communication, Huazhong University of Science and Technology, Wuhan 430074, China; shengzhangboy@163.com (S.Z.); tianyan@hust.edu.cn (Y.T.)

**Keywords:** motion vector, vanishing point, clustering, front-vehicle detection

## Abstract

Assisted driving and unmanned driving have been areas of focus for both industry and academia. Front-vehicle detection technology, a key component of both types of driving, has also attracted great interest from researchers. In this paper, to achieve front-vehicle detection in unmanned or assisted driving, a vision-based, efficient, and fast front-vehicle detection method based on the spatial and temporal characteristics of the front vehicle is proposed. First, a method to extract the motion vector of the front vehicle is put forward based on Oriented FAST and Rotated BRIEF (ORB) and the spatial position constraint. Then, by analyzing the differences between the motion vectors of the vehicle and those of the background, feature points of the vehicle are extracted. Finally, a feature-point clustering method based on a combination of temporal and spatial characteristics are applied to realize front-vehicle detection. The effectiveness of the proposed algorithm is verified using a large number of videos.

## 1. Introduction

In recent years, with the rapid development of intelligent transportation, an increasing amount of attention has been paid to unmanned and assisted driving. As one of the key technologies, vision-based front-vehicle detection has also attracted great attention from both industry and academia. With advantages such as low cost and high performance, vision-based front-vehicle detection can greatly reduce manpower operation and reduce traffic accidents caused by human factors, so it can be effectively applied in assisted or unmanned driving. Facing complex traffic scenes and variable imaging environments, it is undoubtedly extremely challenging to rely solely on vision-based front-vehicle detection to achieve high-precision and high-efficiency objectives. Therefore, it is theoretically significant and has practical application to the study of vision-based front-vehicle detection.

Unlike vehicle detection in static scenes, for the front-vehicle detection problem in unmanned driving the scene changes as the vehicle moves, so the background is not constant and the traditional target-detection method based on background prediction is hardly applicable. At the current stage, a large amount of work remains to be done for front-vehicle detection. In general, vision-based front-vehicle detection algorithms can be divided into the following categories: motion-based methods, feature-based methods, methods based on multi-vision (stereoscopic vision), model-based methods, machine-learning-based methods, and multi-sensor fusion methods.

(1) Motion-based methods

Such methods mainly include the frame-difference method and optical flow method. The frame-difference method [1] is suitable for video monitoring systems with a constant background, whereas its function is rather limited in dynamic road environments. The optical flow method [2] can extract information such as velocity, three-dimensional surface structure, and gray value of the moving targets. However, in some cases susceptible to vibration, the acquired images are usually blurred, resulting in false detection and missed detection of the front vehicle. In addition, the optical flow method is computationally intensive, which makes it difficult to ensure real-time performance of vehicle detection.

(2) Feature-based methods

Also known as knowledge-based methods [3], feature-based methods are the most commonly used methods in vision-based vehicle-detection methods. Common vehicle features include color, texture, headlights, symmetry, lane lines, and underbody shadows. Since the actual scene is often complicated, it is difficult to detect the vehicle only with one kind of feature. Hence, multiple features are often combined in front-vehicle detection. The general process of this method is to first perform a coarse detection of the front vehicle based on specific features, and to then use other features to post-process the coarse-detection results.

Qu et al. [4] proposed a multi-feature front-vehicle detection method that first adopted pre-processing methods, such as graying, smoothing filtering, and histogram equalization to improve the image quality, then used a segmentation algorithm to roughly segment the background and the vehicle in the image, and finally detected the vehicle by comprehensively considering the directional characteristics of underbody shadows and the rear of the vehicle, as well as the symmetry of the vehicle. Similar to the method proposed by Qu et al., [4], Chen et al. [5] proposed to first perform the lane-line detection to determine the approximate position of the vehicle in the image, and then verify the candidate area using constraints like the edge features, vertical projection, and symmetry. By calculating the depth of the road image based on built-in parameters of the camera using the calibration method, Chao et al. [6] determined the difference between the boundary area of the current frame and that of the next frame to obtain the vehicle mask, then obtained the vanishing point through lane-line detection, and finally worked out the position of the vehicle according to the vehicle mask. Li et al. [7] proposed a vehicle detection method based on an And–Or Graph (AOG) and Hybrid Image Templates (HITs). Zhang et al., [8], based on the grads feature of the shadow, which shows the candidate vehicle regions, eliminated the noises of the corresponding area by the method of differential box counting. Then, the accurate vehicle area can be located by analyzing the information of the vehicle’s horizontal edge feature in the candidate vehicle region. Xu et al. [9] proposed an algorithm for front-vehicle detection based on Haar-like features. When the scene is relatively simple, the lighting conditions are good, and the vehicle features are obvious, the above methods can achieve relatively good results. Nevertheless, once the scene becomes complicated or there are a large number of vehicles in the scene, the accuracy will decrease.

(3) Methods based on multi-vision (stereoscopic vision)

Methods based on multi-vision require two vision sensors to simultaneously observe the foreground. Zhao and Zhou [10] proposed a front-vehicle detection method based on binocular vision. Image-processing methods, such as the Canny operator and Hough transform, were first used to detect the road area, then the shadow-recognition method was used to identify the vehicle within the road area, and binocular vision was finally applied to estimate the distance between the unmanned vehicle and front vehicle. Although front-vehicle detection methods based on multi-vision can gain more accurate detection results than the single-vision front-vehicle detection algorithm, the hardware cost of such methods is relatively high, the calculation amount is large, and the matching of features is difficult to carry out.

(4) Model-based methods

The model-based method detects the front vehicle by building a two- or three-dimensional model based on a large amount of vehicle data collected in advance, and then matching the image to be detected with the model [11]. This method is often used jointly with feature-based methods; that is, the potential front vehicle is first determined by feature-based methods, and model-based methods are used for verification to eliminate false targets. Zeng [12] proposed a method based on a rectangular box model and a u-shaped model. This method first established a rectangular frame model for vehicles relatively close to the unmanned vehicle and built a u-shaped model for vehicles relatively far away, and then conducted the final detection according to the dynamic changes of vehicles in front. Pi et al. [13] first used the image gray gradient to detect the front vehicle. Kalman filtering was then used to predict the position of the vehicle in the next frame based on the two-dimensional model of the vehicle and located the vehicle near the predicted position using the edge-projection method. Cai et al. [14] proposed a probabilistic framework combining a scene model with a pattern recognition method for vehicle detection by a stationary camera. Chen et al. [15], aiming at the complex scenes of traffic video surveillance and the characteristics of high real-time requirements, proposed a vehicle detection algorithm based on “divide and conquer” thinking.

This type of method depends too much on the model, and the same model is not applicable to all situations.

(5) Machine learning-based methods

Machine learning-based methods usually need to train the classifier by collecting positive and negative sample sets of the target. The detection process is to input the original image into the trained network and then output the detection results [16]. Yu et al. [17] proposed a front-vehicle detection method applying the Haar feature to the AdaBoost classifier. The integral graph method was used to calculate Haar-like features, which were then applied to the AdaBoost algorithm to select the features and train the classifier. In the end, the trained classifier was used for detection.

Machine learning-based methods have achieved great results in some cases, but these methods require a large number of positive and negative samples, and the algorithm is so complicated that the real-time performance of the detection cannot be guaranteed.

In recent years, deep learning has developed rapidly in the field of computer vision, and there are currently some studies introducing deep learning into front-vehicle detection. Junghwan et al. [18] proposed a front-vehicle detection method based on a convolutional neural network (CNN). This method first obtained the region of interest (ROI) through lane-line and vanishing-point detection, and then determined the candidate region using edge and shadow detection, with the CNN model applied for verification. Zhao et al. [19] put forward a front-vehicle detection method based on a visual attention mechanism and CNN. This method combined a multi-view attention mechanism with a CNN, using the reinforcement learning method to obtain a CNN model based on the visual attention mechanism and then working out the key regions in the image according to the reinforcement learning algorithm. In addition, information entropy was used to measure the classification confidence to further guide the reinforcement learning algorithm. Georgy et al. [20] proposed an adaptive hybrid network that was divided into two levels, one for feature extraction and the other for multi-layer sensing. In the experiment, there were three sensing layers, each of which contained 80 neurons, and the weight of the network was adjusted by a stochastic gradient-descent algorithm. Wang et al. [21] proposed a vehicle detection algorithm based on a multiple feature subspace distribution deep model with online transfer learning.

In the current stage, the main problem of the front-vehicle detection method based on a deep-learning method is that massive samples are needed. Moreover, parameter adjustment of the deep-learning network and the rapid implementation of the algorithm are also difficulties in this respect.

(6) Multi-sensor fusion methods

Considering the influence of weather, road surface, and exterior appearance of the front vehicle during the actual driving process, single-sensor detection—whether it is by vision sensor or radar sensor—will inevitably have some shortcomings, resulting in a decreased detection rate. It is therefore suggested that both radar and vision sensors be used to detect vehicles. Zeng et al. [22] proposed a method to detect the front vehicle using the Harri-like feature classifier based on underbody shadows. In this study, the ROI was obtained using the millimeter-wave radar image, the ROI in the visible image was segmented per the Harri-like rectangular features, and front-vehicle detection was then performed based on the coincidence degree of the ROIs gained by the radar and vision sensors. According to Xie et al. [23], the suspected target was obtained by the video sensor, and the laser distance sensor was used to scan the area. If no beam was reflected back, it would be considered a false alarm, otherwise, the vehicle target would be considered to exist and the distance information would be calculated based on the beam-reflection time. Generally, vision-based vehicle detection methods consist of two stages: hypotheses generation and hypotheses verification. Manuel et al. [24] proposed a feature-based method for on-road vehicle detection in urban traffic. Jin et al. [25] proposed a vehicle detection algorithm combining millimeter-wave radar data and visual multi-features.

Among the above methods, feature-based vehicle detection has been studied quite frequently in the current phase. The research trend has been to extract more effective features and design a reasonable detection algorithm based on features of a front-vehicle detection method. As is known, the characteristics of the front vehicle include the movement velocity characteristics in the time domain, as well as features in the space domain, such as shape, texture, and color. In general, the spatial characteristics utilized in the existing publications are complex and variable, while the movement characteristics are relatively stable. Therefore, with the temporal and spatial characteristics of the front vehicle considered, a motion-vector feature-extraction method based on Oriented FAST and Rotated BRIEF (ORB) and the spatial position constraint was explored in this study. Then, based on the prior knowledge of the vehicle approaching the vanishing point while the background will move away from the vanishing point, the background point was eliminated. In the end, a feature-point clustering method was used to realize front-vehicle detection based on temporal and spatial characteristics. This paper studies how to use spatial characteristics to obtain the motion characteristics of the front vehicle, and how to obtain the final detection of the front vehicle based on the motion characteristics.

## 2. Motion-Vector Extraction Based on ORB and Spatial Position Constraint Matching

The front-vehicle detection method proposed in this study is based on the extraction of a vehicle’s motion vector. The motion vector can be obtained by the optical flow method or inter-frame matching, but the optical flow method is complicated with poor real-time performance. Hence, in this study, an image inter-frame matching algorithm based on ORB [26] and the spatial position constraint was proposed to extract the motion vector.

ORB is a fast feature-extraction and -description method based on the features from accelerated segment test (FAST) feature-point detection [27] and binary robust independent elementary features (BRIEF) [28] description. Since traditional FAST is not scale-invariant, this method introduced a Gaussian pyramid to solve this problem during the extraction of the feature points. During the feature-description stage, based on the BRIEF algorithm, the problem of rotation invariance was solved by introducing the grayscale centroid method.

The feature points of FAST are represented as local maxima and minima points in the neighborhood, and the ORB method is based on FAST, so the ORB feature points also have this feature. For occluded and partial views, as long as the pixel points corresponding to the vehicle have a maximum or minimum value, the corresponding feature points can still be extracted. In other words, for occluded and partial views, FAST feature points have a relatively robust performance. It is worth mentioning that Wang et al. [29] proposed a probabilistic inference framework based on part models to deal with the occlusion and partial-view problems. In addition, viewpoint maps generated by knowledge of road structure are also required. These processes to some extent limit the practical applications of this method.

### 2.1. Algorithm Flow

To obtain the motion vector of the vehicle, it is necessary to match the feature points extracted by the ORB algorithm. However, the common approach cannot guarantee that all feature points can be correctly matched. Moreover, if all feature points are matched one by one, the computational cost will undoubtedly increase. Since the change of the scenes between adjacent frames will not be too large, the spatial position constraint of the candidate matching points was introduced in this study to improve the matching accuracy and reduce the time consumed by the matching process. The basic idea was to open a window centered on the feature point and find candidate matching points in the window. The candidate matching points in the window were ranked from high to low per their weighted similarities, and the candidate matching point with the highest similarity was selected as the matching point. This strategy has the following strengths: First, it reduced the time overhead because it looked for matching points in the neighborhood rather than in the entire picture; second, the matching accuracy was improved by applying the similarity of the features and the spatial position constraint to jointly portray the similarity. Figure 1 shows the above ORB matching method based on the spatial position constraint.

As shown in Figure 1, the ORB algorithm was used to extract feature points as the points to be matched. To determine the similarity of the feature points, the similarity between the feature point p and all candidate matching points in the window was calculated, the results of which were then arranged from large to small, and the candidate point with the highest similarity value was selected as the homonymy point of this feature point.

### 2.2. Matching-Point Search in Opened Windows

p is the point to be matched in the image of the tth frame and $p^{\prime }$ is the pixel point of p in the corresponding position in the image of the t+1th frame. With $p^{\prime }$ as the center, all candidate points in the neighborhood S×S were selected, and their weighted similarity with p was calculated. The candidate matching point q with the highest weighted similarity was selected to be matched with *p*.

In practice, there may be such a situation that the homonymy points in two adjacent frames may not be detected. The following strategies can be adopted: (1) If no feature point is detected in the window, the feature point should be discarded rather than matched; (2) if there are multiple candidate matching points in the window, the candidate point with the highest weighted similarity should be selected for matching.

### 2.3. Calculation of Weighted Similarity

Since the scaling and rotation of two adjacent frames in the video are not obvious, the homonymy points in two adjacent frames will not deviate too much from each other in the image space. Therefore, based on the usual matching method, the spatial position constraint was added to the feature points to be matched to construct the similarity index as follows:(1)$$Dis=\omega_{1}dis_{brief}+\omega_{2}dis_{space},$$
where $dis_{brief}$ is the distance depicted by the BRIEF description operator, $\omega_{1}$ the corresponding weight, $dis_{space}$ the spatial distance, and $\omega_{2}=1-\omega_{1}$ the corresponding weight.

The BRIEF method is a binary descriptor that mainly uses the Hamming distance to measure the similarity between two binary strings. For two binary strings of the same length, the Hamming distance refers to the number of different characters at the corresponding position, which can be expressed by
(2)$$d=\sum_{i=0}^{n}x[i]\wedge y[i],$$
where both x and y are binary string descriptors, n is length, ∧ represents exclusive OR, and d represents the Hamming distance between two binary strings, i.e., the similarity between the feature points to be matched.

According to the principle of the BRIEF algorithm, the value $dis_{brief}$ is an integer between 0 and n, which are generally integers such as 128 and 256. If the length of the pixel of the diagonal is L, then the spatial distance between the two pixels is 0 to L, so the range of the spatial distance should be adjusted to 0 to n. For the sake of simplicity, here [0,L] was directly mapped to [0, *n*] using linear transformation.

## 3. Front-Vehicle Detection Based on Temporal and Spatial Characteristics

Owing to variable weather conditions, complex background, and diverse vehicles, it is difficult to accurately detect the front vehicle with a single feature. A fast and efficient front-vehicle detection method was thus proposed based on a comprehensive consideration of motion characteristics in the temporal domain and proximity characteristics in the spatial domain.

### 3.1. Algorithm Flow

Owing to the perspective projection in the imaging process, a pair of parallel lines in the physical space will intersect at a point on the image plane, which is called the vanishing point. For front-vehicle detection in unmanned driving, the camera is fixed on the currently unmanned vehicle, and from the point of view of the unmanned vehicle all the cars in front will move forward to the vanishing point in the direction of the current vehicle, whereas the background point will move away from the vanishing point in the opposite direction to the current vehicle. Hence, according to the extraction of motion vectors of the feature points introduced in the preceding section, the feature points in the scene can be divided into two categories: the front-vehicle category and the background category. In addition, considering that the motion vectors of the feature points on the same vehicle are almost the same and adjacent to each other, front-vehicle detection can be realized by clustering motions vectors of the feature points. Based on the above analysis, the front-vehicle detection flow shown in Figure 2 is proposed here.

Since images are easily affected by the environment and equipment during acquisition and transmission, the obtained images often contain noise, so it is necessary to perform corresponding pre-processing operations to improve the image quality. Considering the generality of the method, the Gaussian filtering method was used for de-noising and the multi-scale Retinex operator was applied for image enhancement [30]. After the image was pre-processed, the motion vectors were extracted using the inter-frame matching method introduced in the preceding section. The extraction of the vanishing point and feature points of the front vehicle as well as front-vehicle detection were introduced one by one in the following parts.

### 3.2. Extraction of Vanishing Point Based on Automatic Edge Detection and Hough Transform

Owing to the perspective principle of imaging, two parallel lines in a physical space will intersect at a distant point on the image plane, which is the vanishing point.

The lane lines in the image provide a natural way to extract the vanishing points. The lane lines appear as the strong edges in the image, so lane-line extraction can be performed by means of image edge detection and Hough transform. Common edge-detection operators include sobel, prewitt, LoG, and Canny, which require the setting of a threshold. To adapt edge detection to various situations, an edge-detection method based on an adaptive threshold was used in this study [31]. Based on edge extraction, Hough transform and straight-line fitting were used to extract the lane lines, which were then extended to obtain the vanishing point.

In practice, the lane line may be not visible; in this case, we can also use the edges on both sides of the road to extract the vanishing point.

### 3.3. Extraction of Feature Point of Front Vehicle Based on Analysis of Motion Vector

Owing to the relativity of motion, we will feel that the points in the scene are gradually moving away from the vanishing point, while the points on the front car are gradually approaching the vanishing point. It should be noted that when the speed of the front car is greater than that of the current car, the front car will approach the vanishing point; otherwise, the front car will move away from the vanishing point. Therefore, after extracting the vanishing point of the scene in a certain frame, it is fixed for a certain period of time (set as 1/5 s here), so that even if the speed of the front car is less than the speed of the current car, the front car will still approach the vanishing point. Furthermore, in the actual scene, the leaves on the sides of the road may tremble in the breeze. Such movements are rather random, but the movement of feature points on the vehicle is fairly consistent, so the differences between vehicles and the scene in movement velocity can be used to distinguish feature points on the vehicle from those in the background.

As mentioned in Section 2, the ORB feature-point extraction method was used to extract the feature points of the image. Figure 3 shows an example of extraction of leading-vehicle feature points. Subfigure (a1) shows the extracted feature points, including the scenes and the vehicles, and the two points connected by a straight line are homonymy points between the adjacent two frames. Subfigure (b1) shows feature points of the vehicles detected based on motion-vector analysis. It can be seen that feature points on the vehicle can be effectively distinguished from those in the background based on the motion vectors and the vanishing point, thereby facilitating the detection of front vehicles. To more clearly visualize the matching of feature points of different front vehicles in the scene, we display the extraction results of feature points of three front vehicles in the scene, as shown in Figure 4.

As can be seen in Figure 4, feature points of different front vehicles can be well matched, thus guaranteeing the accuracy of motion velocity extraction.

### 3.4. Front-Vehicle Detection Based on Clustering of Spatial Neighborhood Features and Motion-Vector Characteristics

Based on the vanishing point and motion vectors, most of the scene areas can be eliminated. To obtain the position of each vehicle in the image, it is necessary to classify the feature points on the vehicles. Since feature points on different vehicles have different motion vectors, the detection and marking of vehicles can be realized according to the “clustering” of the motion vectors. On this basis, a feature-point clustering algorithm based on motion vector is proposed in this study. The basic flow is shown in Figure 5.

In Figure 5, let n=0 during initialization, the “total number” denotes the number of motion vectors, i.e., the number of feature points of all the vehicles, and the condition to stop clustering is that all motion vectors are processed. Unlike general clustering methods, this algorithm did not set a fixed number of clusters at the beginning. After obtaining the motion vectors, a motion vector was randomly extracted to initialize the clusters, and the motion vector of each feature point was taken out one by one to calculate its similarity with each cluster. If the similarity is greater than a certain value, the motion vector will be classified into this cluster and the cluster center will then be updated. If not, a new cluster will be created with this motion vector as the center. The feature points in the image can be divided into several clusters after going through the above process to traverse all the motion vectors.

Considering that the feature points on the same vehicle are of strong spatial clustering, the similarity can be characterized jointly by the motion vector and spatial distance. $p(x_{t},y_{t})$ and $p(x_{t+1},y_{t+1})$ are the positions of the feature point p in the image at time t and time t+1, respectively, and $r=(x_{t+1}-x_{t},y_{t+1}-y_{t})$ represents the motion vector of p. As shown in Figure 6, $\theta_{x}$ and $\theta_{y}$ indicate the angles between the motion vector and the x and y direction, respectively, and $V=(\theta_{x},\theta_{y},\left\|r\right\|)$ denotes the motion vector of this point. The motion vector of the *k*th feature point can be expressed as
(3)$$V_{k}=(\theta_{kx},\theta_{ky},\left\|r_{k}\right\|),$$
where $\theta_{kx}$ is the angle between the motion vector of the kth feature point and the x direction, $\theta_{ky}$ the angle between the motion vector of kth feature point and the y direction, and $\left\|r_{k}\right\|$ the magnitude of the motion vector of the kth feature point, which was normalized using the following equation:(4)$$\left\|\tilde{r_{k}}\right\|=\left\|r_{k}\right\|/{\left\|r\right\|}_{max},$$
where $\left\|\tilde{r_{k}}\right\|$ is the normalized magnitude of the motion vector, and ${\left\|r\right\|}_{max}=max\left(\left\|r_{i}\right\|\right)$, i=1,2,⋯,N. The normalized motion vector $\tilde{V_{k}}$ can be expressed as
(5)$$\tilde{V_{k}}=(\theta_{kx},\theta_{ky},\left\|\tilde{r_{k}}\right\|)\;.$$

As the correlation coefficient is suitable for calculating the similarity of vectors, the correlation coefficient was used to measure the similarity between motion vectors. The similarity between the kth motion vector $V_{k}$ and the vector of the ith cluster center $C_{i}$ is denoted by $\rho_{ki}$, and the following equation can be obtained:(6)$$\rho_{ki}=\frac{V_{k}C_{i}^{T}}{\left\|V_{k}\right\|\cdot \left\|C_{i}^{T}\right\|},$$
where $C_{i}^{T}$ is the transposed $C_{i}$, and $\left\|V_{k}\right\|$ and $\left\|C_{i}^{T}\right\|$ are the moduli of the vectors $V_{k}$ and $C_{i}^{T}$, respectively. The range of the correlation coefficient $\rho_{ki}$ was thereby obtained, i.e., [−1,1]. The larger the value, the more similar the two vectors are to each other and the more likely it is for $V_{k}$ to be classified into the cluster represented by $C_{i}$.

Since the vehicle is a rigid body and the feature points on it are strongly correlated when it comes to spatial position, spatial constraint can be introduced into the clustering process and the similarity can be jointly characterized by the motion vector and spatial position. The Euclidean distance was used to measure the spatial distance *d* between two feature points; that is,
(7)$$d=\sqrt{{\left(x_{t+1}-x_{t}\right)}^{2}+{\left(y_{t+1}-y_{t}\right)}^{2}}.$$

To be consistent with the motion vector, the spatial distance was also normalized, and the normalized spatial distance was recorded as $\tilde{d}$. For the sake of simplicity, linear normalization was used, and the following equation was obtained:(8)$$\tilde{d}=d/D,$$
where $D=\sqrt{x_{max}^{2}+y_{max}^{2}}$, and $x_{max}$ and $y_{max}$ represent the maximum values of the x and y coordinate of the starting points in all the motion vectors, respectively.

$\rho_{ki}$ is the similarity between the motion vector $V_{k}$ of the kth feature point and the motion of the ith cluster center $C_{i}$, $\tilde{d_{ki}}$ is the spatial similarity between this feature point and the cluster center, and the total similarity is denoted *S*:(9)$$S=\alpha \cdot \rho_{ki}+\left(1-\alpha \right)\cdot d_{ki},$$
where α represents the weight of the similarity between the vectors, and (1−α) indicates the weight of the spatial similarity. The larger S is, the more likely it is that the two feature points are on the same vehicle, and the more they should be classified into the same cluster. It was proven by many tests that when α is to set to 0.6, the results are good.

Through the clustering of motion vectors, feature points with similar motion vectors were clustered into the same category, and different vehicles were thereby marked. As shown in Figure 7, feature points on different vehicles were marked with different colors. It should be pointed out that the feature-point extraction method adopted in this paper generates some false detection when there is a shadow. Therefore, after clustering, these false-detection shadow points will remain as points on the vehicle. Despite the presence of false positives, these false shadows and points on the actual vehicle form the vehicle area. To improve the accuracy of detection, the method of shadow removal should be studied to eliminate the influence of shadow on the extraction of vehicle feature points.

The front-vehicle detection method proposed in this paper is based on the vanishing point. From the viewpoint of the current vehicle, the vehicles in the opposite lane will be far from the vanishing point, and the background point will also be far from the vanishing point, but the speed of the two will be greatly different, so that the vehicles in the opposite lane can be detected. In addition, as visible-light video is used here, the contrast of the scene is very low for the nighttime situation, and it is difficult to carry out effective vehicle detection. Low contrast image enhancement methods can be considered to improve image contrast and image quality.

## 4. Results and Discussion

### 4.1. Experimental Data

Three videos were taken by an on-board camera, which provided daily test data of the unmanned teams at the State Key Laboratory of Information Engineering in Surveying, Mapping and Remote Sensing of Wuhan University, China [26]. The data are highly targeted to unmanned driving scenes and have practical test application value.

### 4.2. Results and Discussion

Figure 8 shows the results obtained using the method proposed in this study. Each box in Figure 8, Figure 9 and Figure 10 is obtained by taking the minimum external rectangular frame for all feature points on the final extracted vehicle.

It can be seen from Figure 8, Figure 9 and Figure 10 that, in most cases, this algorithm can achieve great detection results. Although the complete areas of the vehicles were not obtained at some time points, the presence of the vehicle was detected.

To further test the performance of the algorithm, a car video containing 1058 vehicles was used for testing. Its image resolution was 1920 × 1080, its frame rate 25 frames per second, and the total number of frames 1200. The parameters were set as follows: the weight of the spatial distance $\omega_{2}=0.3$ during inter-frame matching and the weight of the vector similarity α=0.6 in vector clustering. Since feature-based front-vehicle detection was explored in this study, currently available methods with relatively great effects were selected to be compared with the proposed method. As mentioned in the Introduction, the three methods proposed by Qu and Li [4], Chen et al. [5], and Chen et al. [6] are selected as the comparison methods. To verify the effectiveness of the method presented in this paper, we also compared it to the methods in [7,8]. The values set in this study served as the parameters used in the process to implement these methods. The configuration of the computer used in the experiment is as follows: Pentium (R) Dual-Core processor, CPU E6600 @ 3.06 GHz/3.07 GHz, RAM 8.00 GB, and the Windows 7 64-bit operating system. Precision=TP/(TP+FP) and Recall=TP/(TP+FN), where TP is the number of vehicles detected as the targets, FP the number of non-target objects detected as targets (false detection), and FN the number of undetected targets (missed detection). The values of the performance indexes of the above methods are shown in Figure 11, Figure 12 and Figure 13.

As can be seen in Figure 11 and Figure 12, out of all of the method of [4,5,6], method of [5] has the highest recall, but its precision is the lowest. This is because method of [5] adopts the symmetry of vehicle edge features, and many non-vehicle targets, such as buildings, will be mistakenly detected as vehicles. The method proposed in [6] has the highest precision, but its recall is the worst. This method is very sensitive to the environment and is mostly suitable for single-vehicle situations. In terms of comprehensive recall and precision, the performance results in [4] are better than those in [5,6].

Compared with the above three methods, the method proposed in this study has the highest detection precision, which is 1.1% higher than that of the second-ranked method, that proposed by Chen et al. [6]. In terms of recall, the method proposed in this study also has obvious advantages, and is 1.85% higher than that of the second-ranked method, that proposed by Chen et al. [5]. Regarding time consumption, the method proposed in this study is the least time-consuming, which is close to quasi-real-time.

In addition, it can be seen from Figure 11 and Figure 12 that the precision of the method of [7] is 1.09% higher than that of [8], but 1.49% lower than that of this study. The recall of [8] is 2.54% higher than that of [7], but 1.92% lower than that of this study. Therefore, the detection effect of this method has the best performance in both precision and recall. In addition, the efficiency of the algorithm in this paper is the highest.

Partial experimental results of the above methods are shown in Figure 14.

## 5. Conclusions

Front-vehicle detection for unmanned and assisted driving was explored in this study, based on videos of monocular visible light. To obtain the motion vectors of image feature points, an ORB matching method, integrating spatial constraints, was proposed in this study. This method with spatial constraints not only improved the precision of the algorithm, but also reduced its complexity. According to prior knowledge that the front vehicles are approaching the vanishing point while the background is moving away from it, the feature points of the vehicles were successfully extracted. Finally, vehicle detection was completed by a clustering method based on velocity characteristics and spatial proximity characteristics. The feature-based front-vehicle detection method proposed in this study has a good performance in both detection efficiency and effectiveness. The next phase of work is to optimize this algorithm to further enhance its effectiveness and efficiency.

## Figures and Tables

**Figure 1 sensors-19-01728-f001:**
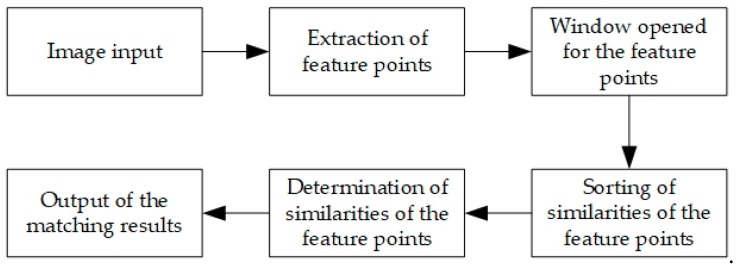
Flowchart of the Oriented FAST and Rotated BRIEF (ORB) matching algorithm based on spatial position constraint.

**Figure 2 sensors-19-01728-f002:**
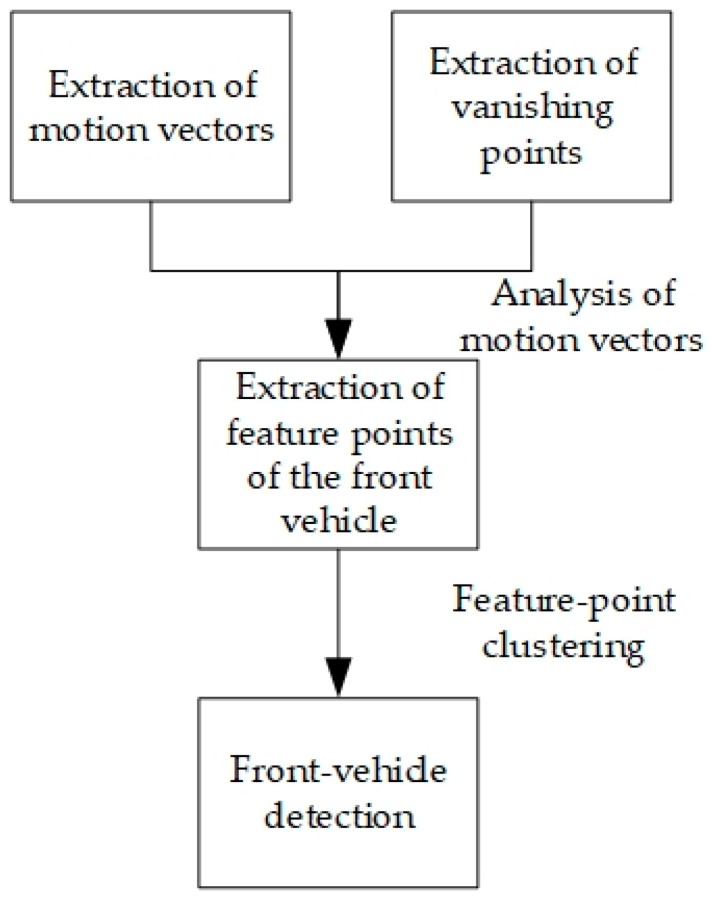
Flow of front-vehicle detection based on clustering of temporal and spatial characteristics.

**Figure 3 sensors-19-01728-f003:**
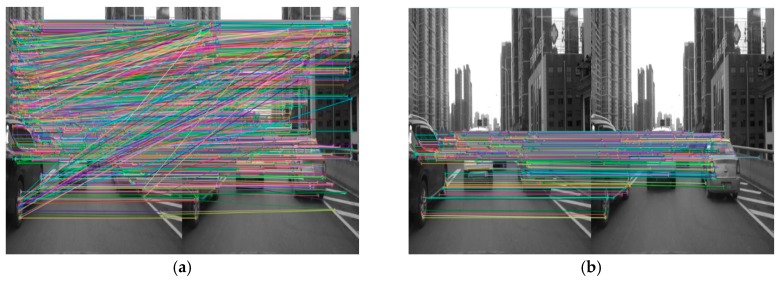
Extraction of feature points of front vehicle based on analysis of motion vectors. (**a**) Matching of feature points in entire image; (**b**) Matching of feature points of front vehicle.

**Figure 4 sensors-19-01728-f004:**
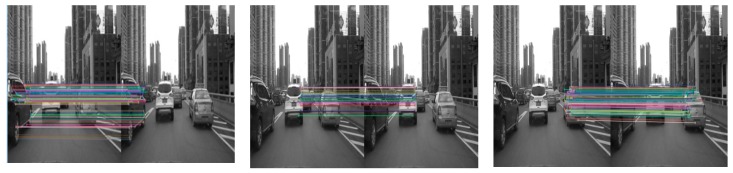
Extraction of feature points of different front vehicles.

**Figure 5 sensors-19-01728-f005:**
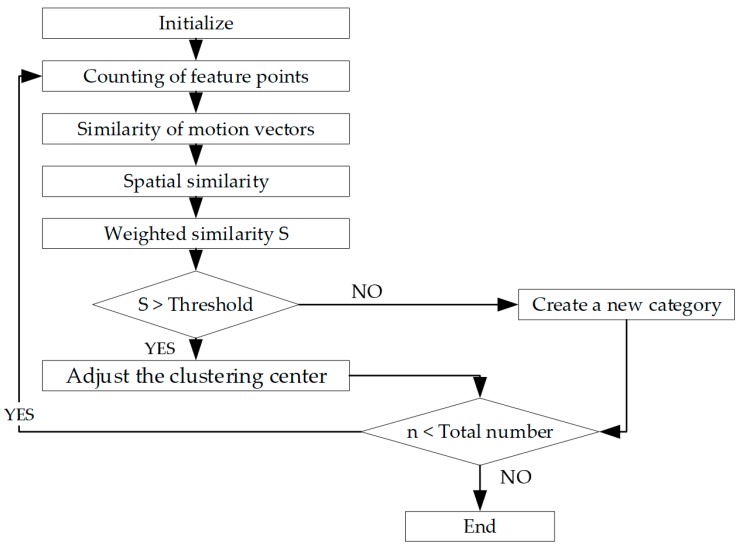
Flowchart of feature-point clustering based on motion vectors.

**Figure 6 sensors-19-01728-f006:**
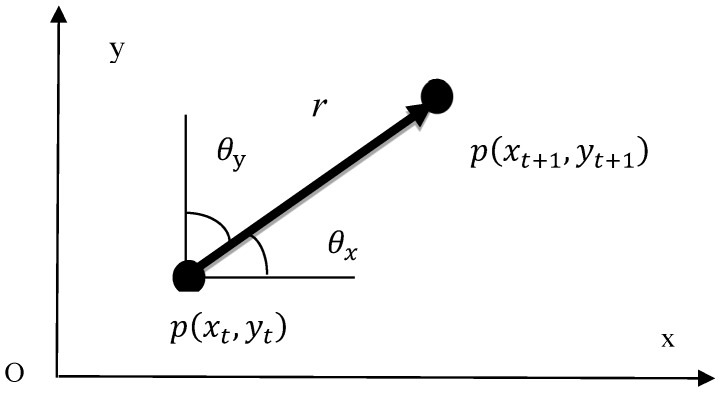
Diagram of motion vectors.

**Figure 7 sensors-19-01728-f007:**
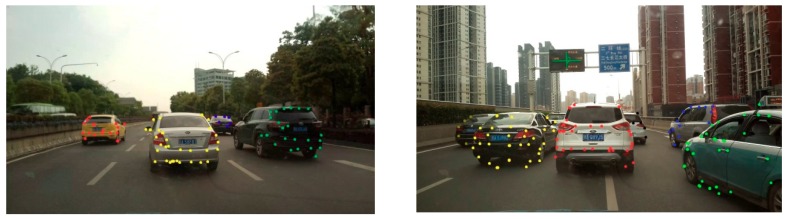
Clustering of motion vectors.

**Figure 8 sensors-19-01728-f008:**
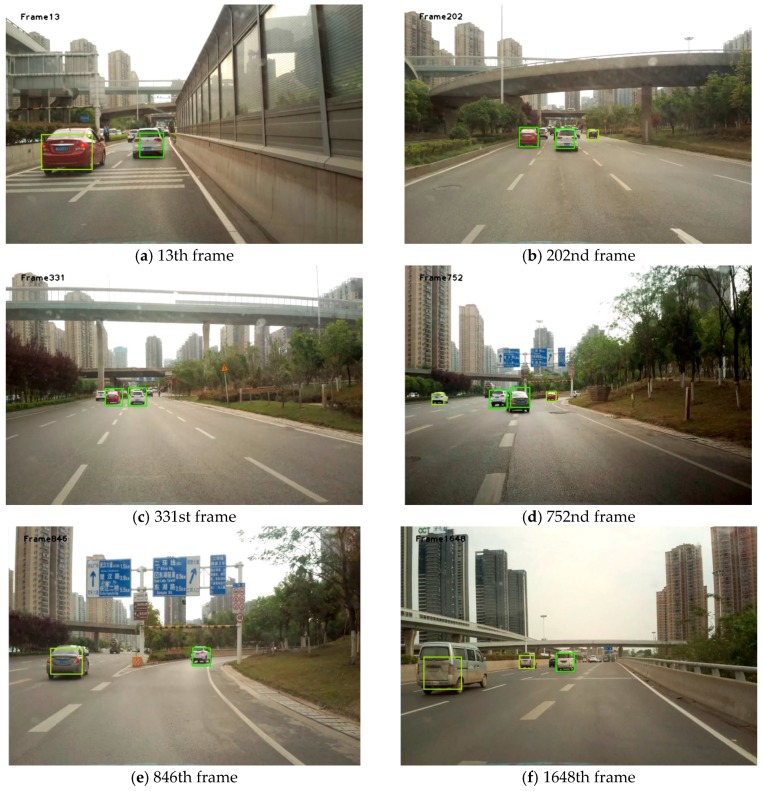
Part of the front-vehicle detection results in Video 1.

**Figure 9 sensors-19-01728-f009:**
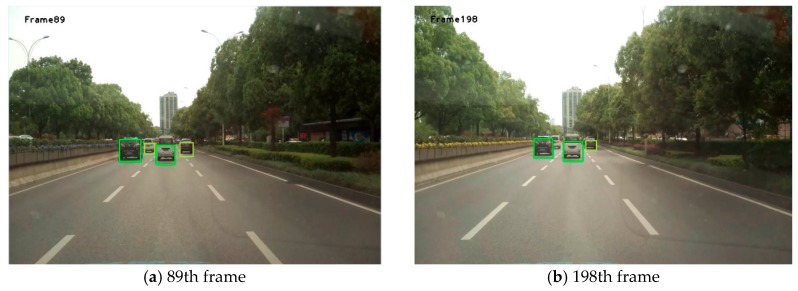
Part of the front-vehicle detection results in Video 2.

**Figure 10 sensors-19-01728-f010:**
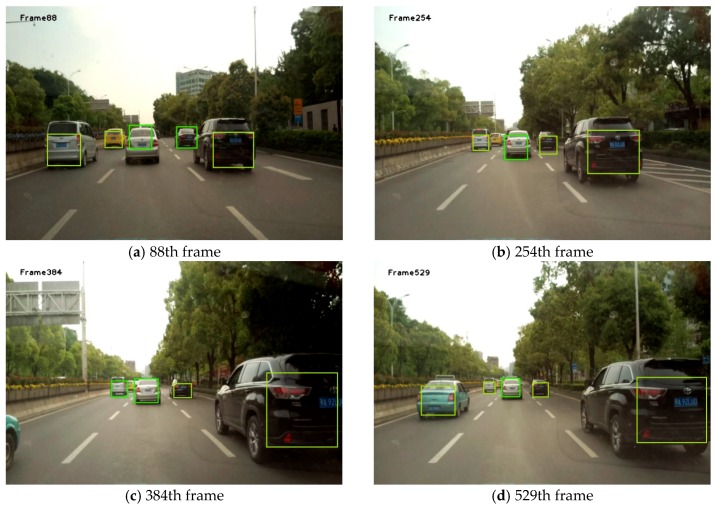
Part of the front-vehicle detection results in Video 3.

**Figure 11 sensors-19-01728-f011:**
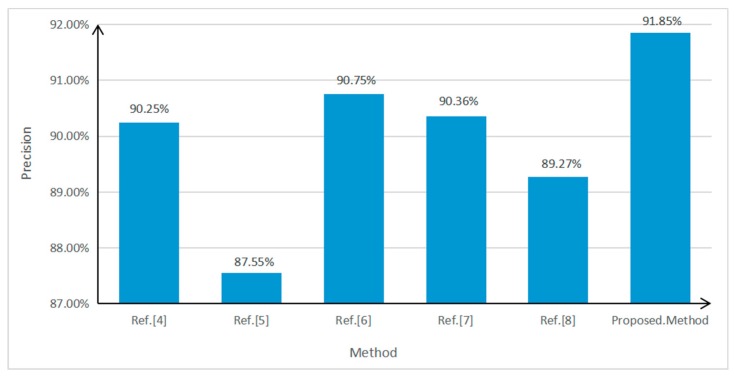
Precision values of various front-vehicle detection methods.

**Figure 12 sensors-19-01728-f012:**
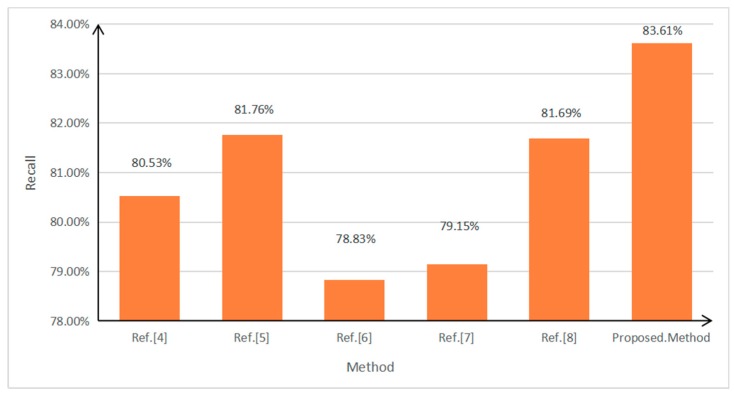
Recall values of various front-vehicle detection methods.

**Figure 13 sensors-19-01728-f013:**
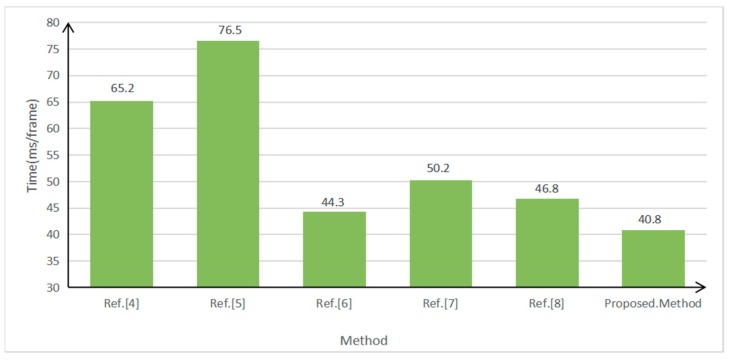
Time consumption of various front-vehicle detection methods.

**Figure 14 sensors-19-01728-f014:**
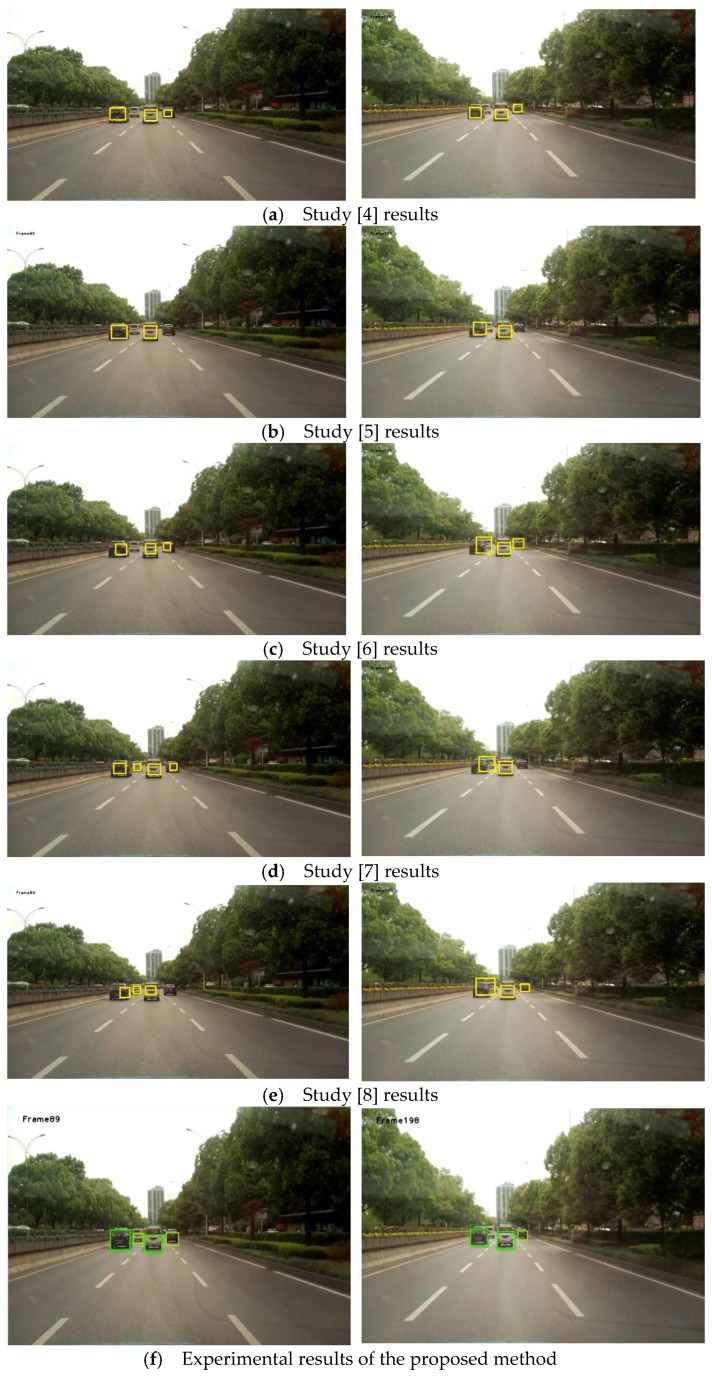
Experimental results of various methods.

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
