# Peer review of "Front-Vehicle Detection in Video Images Based on Temporal and Spatial Characteristics"

_sensors, 2019, doi:10.3390/s19071728_

Reviewer 1 Report

All of my previous comments have been addressed.

Please check the new text highlighted at line numbers 272 - 276, the meaning is clear but the exact wording seem wrong.

Otherwise the paper is ready for publication.

Author Response

Response to Reviewer 1 Comments

Point 1: Please check the new text highlighted at line numbers 272 - 276, the meaning is clear but the exact wording seem wrong. 

Response 1: Many thanks to reviewers for their hard work. As for the problems mentioned by the reviewer in line 272-276, due to the translation problems, the explanation is not clear. We have corrected them. Please refer to line 282-286 of this revised manuscript.

Reviewer 2 Report

First of all, I would like to highlight the effort made by the authors in order to fulfill all my previous comments and I would also want to acknowledge their kindness by delivering the responses.

In the responses to previous review the Authors have explained that:

" Since the vanishing point corresponds to the focus of two parallel lines on the road direction, if the lane line is not visible, the vanishing point can also be obtained if the edges on both sides of the road can be extracted. "

" The front vehicle detection in this paper is based on the vanishing point. Under the view point of the current vehicle, it is difficult to extract the vanishing point in the non-current lane, so the method in this paper is difficult to extract the vehicle on it.In addition, as the visible light video is used here, the contrast of the scene is very low for the night situation, and it is difficult to carry out effective vehicle detection. "

The above shortcomings of the method should be mentioned in the paper. The Authors could also indicate potential approaches to alleviate these shortcomings.

Moreover, the issues related to occlusion and partial view should be discussed. The Authors should explain how their method deals with the occlusion and partial view issues. In this context the survey of related works and state-of-the-art should be extended (see Wang, C., Fang, Y., Zhao, H., Guo, C., Mita, S., & Zha, H. (2016). Probabilistic inference for occluded and multiview on-road vehicle detection. IEEE Transactions on Intelligent Transportation Systems, 17(1), 215-229.)

In Authors' response it was explained that "The box in figure 9 is obtained by taking the minimum external rectangular frame for all feature points on the final extracted vehicle". This explanation should be also provided in the paper.

Equations (3) and (5) are identical. Please use different symbols for the non-normalized and normalized vectors.

Formula (8) is not correct. It would be correct for a programming language but not for mathematical expression. As I understand this is not recurrent formula, thus the same symbol (d) cannot appear on both sides of the equation .

The Authors should use box and whiskers plots (or alternatively  provide information about standard deviations) for the precision and recall values shown in Tables 1 and 2. Such modification will improve presentation of the results and will highlight the statistical significance of the differences between precision and recall obtained for the compared methods.

The manuscript needs careful language editing. Below are examples of the language issues:

Abstract: "In this paper, to achieve front-vehicle detection in 14 unmanned or assisted driving, a vision-based, efficient, and fast front-vehicle detection method 15 based on the spatial and temporal characteristics of the front vehicle is proposed in this study." This sentence should be shortened. The last part " in this study " is redundant.

Section 1: " In general, the spatial characteristics utilized in the existing publications are complex and variable, Therefore, the accuracy and robustness of front vehicle detection are improved, while the movement characteristics 160 are relatively stable, which is beneficial to improving the effect of front vehicle detection. ". This sentence is grammatically incorrect.

" This paper studies how to use spatial characteristics for registration to obtain the motion characteristics of the front vehicle, and  how to classify and cluster based on the motion characteristics to obtain the final detection of the  front vehicle. " This sentence is unclear. The part "for registration" in my opinion is not necessary.  What is clustered?

Author Response

Response to Reviewer 2 Comments

Point 1:In the responses to previous review the Authors have explained that:

" Since the vanishing point corresponds to the focus of two parallel lines on the road direction, if the lane line is not visible, the vanishing point can also be obtained if the edges on both sides of the road can be extracted. "

" The front vehicle detection in this paper is based on the vanishing point. Under the view point of the current vehicle, it is difficult to extract the vanishing point in the non-current lane, so the method in this paper is difficult to extract the vehicle on it. In addition, as the visible light video is used here, the contrast of the scene is very low for the night situation, and it is difficult to carry out effective vehicle detection. "

The above shortcomings of the method should be mentioned in the paper. The Authors could also indicate potential approaches to alleviate these shortcomings.

Response 1: Many thanks to the reviewers for their good advice.

For "Since the vanishing point teach to the focus of two parallel lines on the road direction, if the lane line is not visible, Extracted "the vanishing point can also be obtained if the edges on both sides of the road can be extracted."

For the statement "the front vehicle detection in this paper is based on the vanishing point. Under the view point of the current vehicle, it is difficult to extract the vanishing point in the non-current lane, so the method in this paper is difficult to extract the vehicle on it. In addition, as the visible light video is used here, The contrast of the scene is very low for the night situation, And it is difficult to carry out effective vehicle detection "teach to the question" Does the method enable detection of exotic vehicles from the opposite direction in two-directional road With two traffic lanes?". We failed to understand the problem correctly last time. This time we explained the problem again and added it in the revised version. Please refer to lines 371-377 in the revised version.

Point 2: Moreover, the issues related to occlusion and partial view should be discussed. The Authors should explain how their method deals with the occlusion and partial view issues. In this context the survey of related works and state-of-the-art should be extended (see Wang, C., Fang, Y., Zhao, H., Guo, C., Mita, S., & Zha, H. (2016). Probabilistic inference for occluded and multiview on-road vehicle detection. IEEE Transactions on Intelligent Transportation Systems, 17(1), 215-229.)

Response 2: As for the issue of occlusion and partial view, the key is accurate target tracking. The feature point extraction method (ORB) adopted in this paper can improve the tracking effect in both cases. In fact, figure 7 also illustrates the effect of the algorithm in this paper. For the simple analysis of the ORB algorithm, we have added in this revision (180-188).

Point 3: In Authors' response it was explained that "The box in figure 9 is obtained by taking the minimum external rectangular frame for all feature points on the final extracted vehicle". This explanation should be also provided in the paper.

Response 3: This section has been added to the revised text as shown in lines 385-387.

Point 4: Equations (3) and (5) are identical. Please use different symbols for the non-normalized and normalized vectors.

Response 4: We use (please see in attachment) and to denote the non-normalized and normalized vectors in the modified version.

Point 5: Formula (8) is not correct. It would be correct for a programming language but not for mathematical expression. As I understand this is not recurrent formula, thus the same symbol (d) cannot appear on both sides of the equation.

Response 5: Formula (8) is (please see in attachment),denotes the normalized Euclidean distance and (please see in attachment) is the Euclidean distance between two feature point, they are different.

Point 6: The Authors should use box and whiskers plots (or alternatively  provide information about standard deviations) for the precision and recall values shown in Tables 1 and 2. Such modification will improve presentation of the results and will highlight the statistical significance of the differences between precision and recall obtained for the compared methods.

Response 6: Thank the reviewers for their Suggestions.The results in table 1 and table 2 were modified according to the Suggestions of reviewers, and the corresponding figure number and description were modified.

Point 7: The manuscript needs careful language editing. Below are examples of the language issues:

Abstract: "In this paper, to achieve front-vehicle detection in 14 unmanned or assisted driving, a vision-based, efficient, and fast front-vehicle detection method 15 based on the spatial and temporal characteristics of the front vehicle is proposed in this study." This sentence should be shortened. The last part " in this study " is redundant.

Section 1: " In general, the spatial characteristics utilized in the existing publications are complex and variable, Therefore, the accuracy and robustness of front vehicle detection are improved, while the movement characteristics 160 are relatively stable, which is beneficial to improving the effect of front vehicle detection. ". This sentence is grammatically incorrect.

" This paper studies how to use spatial characteristics for registration to obtain the motion characteristics of the front vehicle, and how to classify and cluster based on the motion characteristics to obtain the final detection of the front vehicle. " This sentence is unclear. The part "for registration" in my opinion is not necessary. What is clustered?

Response 7: Thanks to the careful review of the reviewers, we have corrected the above language problems.

Round  2

Reviewer 2 Report

The paper needs text editing but can be considered for acceptance.

This manuscript is a resubmission of an earlier submission. The following is a list of the peer review reports and author responses from that submission.

Round  1

Reviewer 1 Report

The paper makes a worthwhile contribution to the topic of camera-based tracking, very specific to the detection and tracking of vehicles in front of a subject vehicle. The introduction gives a fair and clear description of the state of the art. There are a number of improvements that can be made regarding explanation and presentation, all minor:

line 262: the points of the front vehicle may or may not approach the vanishing point, depending on the relative motion of the two vehicles - please clarify.

Figure 3. There is no need to show many images with image-wide feature matches, (a1) vs (b1) is sufficient. Further examples of front vehicle matches are worthwhile, and in at least one case the authors should (manually) separate the matching pairs to the level of the single vehicle. In this way the reader can see more clearly the type of features being matched.

It appears that in Figure 6, the images were captured from a stationary camera. This should be clarified and at least one image should show clustering from images captured in a moving vehicle.

Reviewer 2 Report

The paper deals with front-vehicle detection in video sequences from on-board cameras. This detection task is important for development of safety systems and unnamed vehicles. It should be noted that the research topic is of current interest and number of methods are available in the related literature. The manuscript needs several improvements.

The authors should better explain novelty of their proposed approach in context of the existing methods, discussed in Sect. 1. Limitations of the existing methods should be indicated that motivate the development of the new approach.

The following terms are unclear: "vanishing point", background was moving away from the vanishing point", "background is moving away from the vanishing point in the opposite direction", "cars in front are moving forward in the direction of the vanishing point".

Additional explanations are also needed with regard to the extraction of feature point of front vehicle based on analysis of motion vector. It is not clear how the feature points of front vehicle are selected. Detailed algorithm should be presented. Moreover, extended discussion of the results presented in Fig. 3 is required. In this figure it is definitely not obvious how the feature points on the vehicle can be effectively distinguished from those in the background.

Is it possible to extract the vanishing point in case when the lane lines are not visible?

Does the method enable detection of approaching vehicles from the opposite direction in two-directional road with two traffic lanes? How the method performs in night conditions? It would be useful to present examples of detection results for such situations.

The flowchart in Fig. 4 is incorrect. For the decision "n<Total number" an output arrow with label "YES" is missing.

Equations (3) and (5) are identical.

Formula (8) is not correct.

It should be discussed how to deal with the false detections of feature points in shadowed regions (see the bus in Fig. 6 left).

How the boxes in Fig. 9 were determined?

Authors should provide additional analysis of the performance results (e. g. by using box plots) to show if the limited differences among methods in Table 1 are statistically significant.

The following sentences are confusing as the highest recall and precision in Tab. 1 are presented for the proposed method: "As can be seen in Table 1, the method of Refs. [4]–[6] has the highest recall, " vehicles. The method proposed in [6] has the highest precision, but the recall is the worst"

The compared methods (Refs. [4] - [6]) are mainly from conference proceedings, thus they are not fully representative for the state-of-the-art. It would be interesting to see comparison of the results achieved by the proposed method with the other methods published in highly-ranked journals (as the Sensors journal) or e.g., [7] and [8].